# Omni TM-AE: A Scalable and Interpretable Embedding Model Using the Full Tsetlin Machine State Space

## Abstract

The increasing complexity of large-scale language models has amplified concerns regarding their interpretability and reusability. While traditional embedding models like Word2Vec and GloVe offer scalability, they lack transparency and often behave as black boxes. Conversely, interpretable models such as the Tsetlin Machine (TM) have shown promise in constructing explainable learning systems, though they previously faced limitations in scalability and reusability. In this paper, we introduce Omni Tsetlin Machine Autoencoder (Omni TM-AE), a novel embedding model that fully exploits the information contained in the TM's state matrix, including literals previously excluded from clause formation. This method enables the construction of reusable, interpretable embeddings through a single training phase. Extensive experiments across semantic similarity, sentiment classification, and document clustering tasks show that Omni TM-AE performs competitively with and often surpasses mainstream embedding models. These results demonstrate that it is possible to balance performance, scalability, and interpretability in modern Natural Language Processing (NLP) systems without resorting to opaque architectures.

## 1 Introduction

Artificial intelligence models in the field of Natural Language Processing (NLP) have recently delivered impressive capabilities that have significantly transformed the delivery of web-based services. This surge in development has intensified competition to produce increasingly powerful models, prompting the creation of larger and more complex architectures. While this momentum has driven the provision of enhanced services, such as improved customer interactions on the web, it has also resulted in the neglect of a crucial concern: the ability to explain and analyze model behavior, such as understanding why a particular response is selected over others. This complexity originates from the fundamental building blocks of these models. For instance, during the token embedding stage, the relationships between tokens are encoded through mathematical constructs derived after extensive training, resulting in dense, multi-dimensional float vector representations. Such an information structure makes it virtually impossible to trace decision-making processes, particularly when embeddings are embedded in larger architectures like Large Language Models (LLMs).

Tsetlin Machine (TM), a machine learning model employing logical decision-making processes, introduced a novel approach for pattern recognition and decision-making that mimics artificial intelligence while remaining interpretable and explainable (Granmo, 2018). The initial embedding model, proposed in Bhattarai et al. (2024), demonstrated success in similarity calculations but faced scalability challenges beyond relatively limited datasets, particularly in applications like clustering that demand scalable solutions. The scalability limitation arose because training was conducted on a full input vector: to compute similarities for a set of words, the model required simultaneous input of all words in a single vector. For instance, if the model was trained to compute similarity for a vector containing three tokens $x_1$, $x_2$, and $x_3$, the resulting trained model would be tightly bound to this specific set of tokens $\{x_1, x_2, x_3\}$. Consequently, if the vector contents were subsequently modified, e.g., through the addition, deletion, or substitution of any token, the output would become invalid, necessitating complete retraining.

Kadhim et al. (2025) sought to address these limitations by proposing a scalable model that trained tokens individually rather than collectively. In that approach, each token within a vector could be trained separately, and context information for each could be independently collected. However, although that method succeeded in overcoming the initial scalability issue, it introduced new challenges. Specifically, the context information collected during the first training phase was insufficient for direct use in more complex tasks such as similarity measurement or clustering. If one wished, for example, to compute the similarity between two separately trained tokens, such as $x1$ and $x2$, it was necessary to undergo a second training phase to establish relationships between them. This reliance on additional retraining significantly increased the model's complexity and training time, and limited its practicality for real-world applications.

In this work, we propose a novel method for generating reusable embeddings using only a single training phase, eliminating the need for retraining. In addition, to the best of our knowledge, this is the first approach to fully exploit all literals within the TM clause, including literals that are often excluded. Specifically, we utilize literals below a certain threshold $N$, representing literals that can reliably contribute to describing target classes. By incorporating this previously excluded information, our models demonstrate substantial improvements in measuring similarity, classification, and clustering across NLP applications. Furthermore, training time is greatly reduced by removing the dependency on additional training stages, thereby enhancing the model's reusability. Our contributions are based on the following three key aspects:

1. Introducing the first TM model that utilizes the information from excluded literals, thus improving embedding quality. We refer it to as "Omni TM-AE".

2. Enabling the computation of embeddings from a single phase, without the need for retraining, and in a reusable manner.

3. Presenting the first propositional logic-based embedding model whose information can be effectively applied across various NLP tasks, with comparable performance to deep learning based models such as ELMo and BERT, etc.

## 2 Related Work

The embedding phase constitutes one of the foundational steps in most LLMs, including architectures such as Transformers (Vaswani et al., 2017). Early embedding models, such as GloVe (Pennington et al., 2014), introduced simple statistical methods for constructing word embeddings by counting the frequency of word occurrences within the training dataset. Word2Vec (Mikolov et al., 2013) and its improved version FastText (Bojanowski et al., 2017) advanced this field by providing a unique representational space for symbols through the collection of contextual information from training data. These models successfully replaced earlier statistical NLP methods for embedding construction and paved the way for the development of large language models by integrating embeddings at multiple stages of their architecture. However, these models inherently lack interpretability and explainability, and they treat tokens as mathematical entities within a black-box framework trained solely to extract contextual features from datasets.

TM represents a distinct paradigm in machine learning, characterized by its transparent and interpretable structure (Granmo, 2018). It employs logical expressions to construct patterns that reveal the underlying classes they describe (Glimsdal & Granmo, 2021; Kadhim et al., 2024; Yadav et al., 2022; Abeyrathna et al., 2023; Sharma et al., 2023). The TM has shown strong performance across a wide range of applications (Abeyrathna et al., 2021), including aspect-based sentiment analysis (Yadav et al., 2021a), text classification (Yadav et al., 2021b; 2022), contextual bandit problems (Seraj et al., 2022), image classification (Grønningsæter et al., 2024; Jeeru et al., 2025), and federated learning (Qi et al., 2025). In addition, TMs are inherently hardware-friendly and exhibit high energy efficiency when implemented on hardware (Tunheim et al., 2025a; Maheshwari et al., 2023; Tunheim et al., 2025b), making them especially well-suited for IoT devices and edge computing scenarios where low power usage and computational efficiency are critical. While real-world deployment can introduce some variability, convergence analyses (Jiao et al., 2023; Zhang et al., 2022) demonstrate that TMs almost surely converge to fundamental Boolean operations, underscoring the robustness.

In the field of NLP, the implementation of TM as described by Bhattarai et al. (2024) marked its first use in generating word embeddings, known as Tsetlin Machine Autoencoder (TM-AE) that

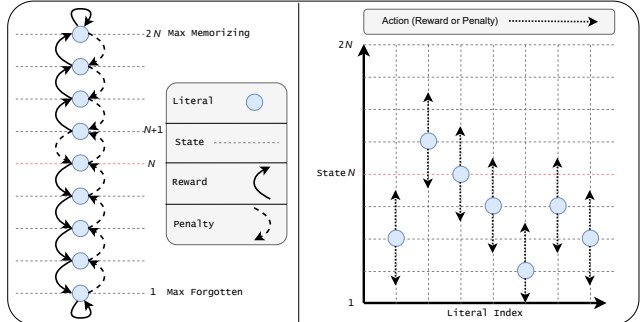

Figure 1: (Left) State-transition diagram for a single Tsetlin automaton, illustrating penalty/reward actions between the ground state=1 "max-forgotten" to the $2N$ "max-memorizing" state. (Right) Clause construction space: the $x$-axis indexes literals (vocabulary features and their negations) while the $y$-axis records their current automaton state. Literals above threshold $N$ are selected to constitute the clause.

facilitate understanding the reasoning behind specific output results. However, a major limitation was the difficulty in reusing the training outputs or extending them to accommodate larger input vectors. In Kadhim et al. (2025), an attempt was made to solve the problem of reusing context information. However, because that method relied heavily on the final clauses and the layers contained within it, the extracted information is usually insufficient to build embeddings through a single phase. Instead, the process required two distinct stages, thereby impeding the model's ability to achieve rapid embeddings for complex applications such as clustering. In the present work, we propose a novel TM-based autoencoder to construct a reusable and scalable single-stage embedding.

# 3 THE TSETLIN MACHINE AND EMBEDDING ARCHITECTURE

## 3.1 TSETLIN AUTOMATA

The TM architecture comprises a collection of learning units referred to as Tsetlin automaton (TA). Each automaton undergoes state transitions based on reinforcement learning, either rewards or penalties, determined by its evaluation. Figure 1 (left) illustrates the state transition dynamics of an automaton for a literal. In this context, a $literal$ denotes either a feature from the vocabulary or its negation. Consequently, for a vocabulary of size $d$, the total number of literals is $2d$.

The vertical axis in the figure represents the automaton's state, which ranges from complete forgetting (state 1) to maximal memorization (state $2N$). The higher a literal's state, the more likely it is to be retained for clause formation. Literals that frequently contribute to accurate classification tend to ascend the state space, whereas less relevant literals tend to descend.

## 3.2 CLAUSES

To construct clauses, which are logical expressions used for classification, TA are organized into groups encompassing all literals (see Figure 1, right). A clause is a conjunction (logical AND) of literals selectively chosen based on their state within the TA. The figure illustrates clause composition, where the $x$-axis denotes literal indices and the $y$-axis corresponds to TA states of literals. Literals above a defined threshold $N$ are selected to construct the clause, reflecting their strong association with the target class. For instance, when training a class (word) labeled "happy," a resulting clause may take the form of "marriage AND love" or "family AND home", capturing contextual semantic relationships through logical expressions.

## 3.3 AUTOENCODER AND COALESCED MECHANISM

The TM-AE employs a specialized structure known as Coalesced Tsetlin Machine (CoTM), which enables multi-output learning by training multiple target classes in parallel. The process, as shown in Figure 2, comprises three main stages:

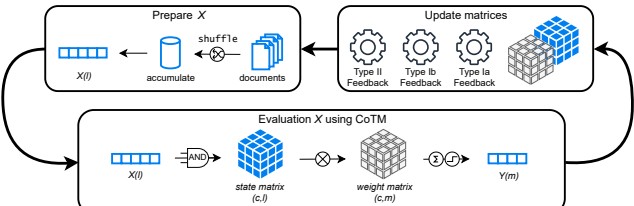

Figure 2: In The TM-AE structure the process of each example in epochs consists of three phases: (1) Preparation, where input vectors are generated from documents and encoded into binary form. (2) Evaluation, where input $X$ is processed through a state matrix, weight matrix, and logical conditions to determine output predictions. and (3) Update, where TAs in a clause adjust their states to include or exclude literals based on whether the clause matches the input data.

1. **Preparation Phase:** The input vector $X$, encoded in binary form, encapsulates document-level information. Its length $l$ is twice the vocabulary size, given by $l = 2d$, to account for features and their negations. Training instances labeled with 1 are derived from documents containing the target class, whereas those with output 0 come from documents lacking that class. Documents are shuffled, and a selection is governed by the hyperparameter *accumulation*. Features present in the document are encoded as 1, and their negations as 0, and vice versa.

2. **Evaluation Phase:** This stage assesses if the input sample $X$ satisfies the logical expression of each clause. Clauses are formed by included literals, namely, the literals whose corresponding TAs have states exceeding the threshold $N$. Users can specify an arbitrary number of *clauses*, stored in a two-dimensional matrix $state\_matrix$ with dimensions $c \times l$, where $c$ is the number of clauses and $l$ is the number of literals. The output is weighted using a $weight\_matrix$ with dimensions $c \times m$, where $m$ represents the number of classes. The final prediction $Y$ is derived through majority voting.

3. **Update Phase:** Clause updates are conducted based on the relationship between the input $X$, the clause (via its included literals), and the target output (the label). During training, the TA state within a clause can increase or decrease, effectively determining whether its corresponding literal is included in the clause. In essence, three types of updates—Type Ia Feedback, Type Ib Feedback, and Type II Feedback—are applied to the state and weight matrices. Additional information can be found in Glimsdal & Granmo (2021).

## 3.4 OMNI EMBEDDING MECHANISM

To the best of our knowledge, all prior research in TM has utilized the concept of a clause to characterize the target class by focusing exclusively on literals whose TAs surpass state $N$. In other words, only the included literals in a clause contribute to the final classification. In contrast, the present study leverages the complete information encapsulated within the $state\_matrix$, i.e., not only the included literals, but also the excluded literals. Although the excluded literals do not directly contribute to the final clause expression, they are nevertheless subject to training and may contain important information of the target class, which can be used for embedding.

To demonstrate the existence of useful information in the excluded literals, Figure 3 illustrates TAs inside a clause for a specific target class, "happy". Prior methods have focused on such

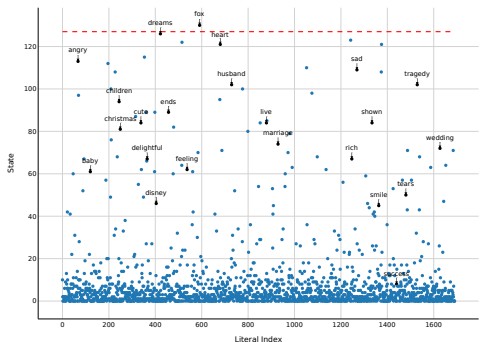

Figure 3: Visualization of a trained clause for the target class "happy," where the word "fox" is the only literal exceeding the threshold state ($N = 128$ red line). Additional literals are present at lower levels and provide latent contextual information.

clauses by considering only the included literals, i.e., whose TA states exceeding state $N$. Following this concept, the particular clause in Figure 3 has a single literal, i.e., "fox", that exceeded the $N$ state (the red line where $N = 128$ in this experiment). However, by observing Figure 3, it is evident that the excluded literals contain substantial informational content (see other annotated words in Figure 3 below state $N$, like "sad", "angry" and "rich"). These excluded literals also provide crucial cues for constructing an informative embedding vector.

In our proposed method, we introduce a novel technique for deriving the embedding vector grounded in the structure of the $state\_matrix$. Using this method, the states of all literals—original features and their negation—including the excluded ones, contribute to the formation of the embedding vector. To build the embedding for a target word, the average of the signed-state values is computed for each feature in the vocabulary by adding the values for the original feature and subtracting those for its negation.

The embedding vector $\mathbf{e} \in \mathbb{Z}^d$ for a certain target word[1] is derived by training TM-AE for the target word and aggregating signed state values associated with each feature across all positive weighted clauses. Denote $\mathcal{D} = \{x_1, x_2, \ldots, x_d\}$ the set of original features (e.g., vocabulary tokens), and define a *literal* as either a feature $x_i$ or its negation $\neg x_i$. Thus, the total number of distinct literals is $2d$.

Let $\mathcal{C}$ be the set of all positive clauses (clause with positive weights) for the target word, and $S(c)$ denote the set of literal-state pairs $(l, n)$ within clause $c \in \mathcal{C}$, where $l \in \{x_1, \neg x_1, \ldots, x_d, \neg x_d\}$ is a literal and $n \in \{1, 2, \ldots, 2N\}$ is the automaton state assigned to $l$.

For this target word, the embedding vector will be of the size of the vocabulary $d$, and for each feature $x_i$ inside it where index $i \in \{1, \ldots, d\}$, the signed state sum $v_i$ is computed as:

$$v_i = \sum_{c \in \mathcal{C}} \left( \sum_{(l,n) \in S(c): l = x_i} n - \sum_{(l,n) \in S(c): l = \neg x_i} n \right),$$

The occurrence count $t_i$ is defined as the total number of positive clauses. Based on these, the embedding component for each $x_i$ is computed as:

$$e_i = \begin{cases} \left\lfloor \dfrac{v_i}{t_i} \right\rfloor & \text{if } t_i > 0, \\ 0 & \text{otherwise.} \end{cases}$$

Thus, the complete embedding vector for the target word is given by:

$$\mathbf{e} = [e_1, e_2, \ldots, e_d].$$

In summary, for each feature, the method consistently considers both the original literal and its negation by adding the state values of the feature and subtracting the state values of its negation. These contributions are combined and normalized by the total number of their clauses, ensuring that the embedding component captures the net influence of the feature in both positive and negative contexts. This approach guarantees that both aspects are always incorporated, providing a balanced and interpretable representation directly derived from the clause structure. For further details on the complete algorithm, please refer to the appendix.

## 4 RESULTS

### 4.1 EXPERIMENTAL SETUP

The proposed model's effectiveness and the quality of the generated embeddings were assessed through four tasks: similarity analysis, classification, visual clustering, and numeric clustering, using publicly available datasets and open-source embedding models for comparison. All experiments were executed in two containerized Ubuntu environments with the CUDA toolkit, running on an NVIDIA

---

[1]The notation displays only the embedding of a single target word. This approach is applicable to compute the embedding of any target word in the same way.

Table 1: Comparison of Spearman (S) and Kendall (K) similarity measures across different datasets and models. Note: TM-AE results for datasets other than RG65 were not conducted due to time constraints.

| Dataset | Word2Vec | | Fast-Text | | GloVe | | TM-AE | | Omni TM-AE | |
|---|---|---|---|---|---|---|---|---|---|---|
| | S | K | S | K | S | K | S | K | S | K |
| WS-353 | 0.587 | 0.410 | 0.600 | 0.420 | 0.504 | 0.347 | N/A | N/A | 0.485 | 0.345 |
| MTURK287 | 0.560 | 0.389 | 0.578 | 0.403 | 0.513 | 0.357 | N/A | N/A | 0.525 | 0.374 |
| MTURK717 | 0.470 | 0.322 | 0.488 | 0.336 | 0.460 | 0.317 | N/A | N/A | 0.479 | 0.343 |
| RG65 | 0.544 | 0.366 | 0.502 | 0.344 | 0.522 | 0.357 | 0.506 | 0.361 | 0.644 | 0.462 |
| MEN | 0.562 | 0.391 | 0.580 | 0.406 | 0.559 | 0.389 | N/A | N/A | 0.597 | 0.440 |
| Avg. | 0.545 | 0.376 | 0.550 | 0.382 | 0.512 | 0.353 | N/A | N/A | 0.546 | 0.393 |

DGX H100 server restricted to two GPUs and with 2.0 TiB of RAM. Computational demands varied by experiment but were consistently managed within this setup. Three Python environments were used: one dedicated to the ELMo model requiring a specific TensorFlow version, another configured with RAPIDS cuML for GPU-accelerated clustering and efficient KMeans computations, and a default environment for the remaining experiments.

## 4.2 Semantic Similarity Evaluation

In this evaluation, the embedding vectors generated by the proposed model were used to compute the semantic similarity between pairs of words for which human-annotated similarity scores are available. Two statistical methods were employed to measure the alignment between model predictions and human judgments. The first method, Spearman's rank correlation, is a non-parametric metric that assesses the strength and direction of a monotonic relationship between two variables. The second method, Kendall's rank correlation coefficient, evaluates the ordinal association between two measured quantities and is also non-parametric in nature.

This experiment utilized five benchmark datasets commonly used for evaluating semantic similarity and relatedness: WS-353, MTURK287, MTURK717, RG65, and MEN. These datasets comprise word pairs along with human-assigned similarity ratings. By comparing the model-generated similarity scores with those provided by humans, the model's effectiveness in capturing semantic similarity can be quantitatively assessed. The datasets varied in size, with RG65 being the smallest (65 word pairs) and MEN the largest (3,000 word pairs).

For comparative analysis, five embedding models were selected: Word2Vec, FastText, GloVe, TM-AE, and the proposed Omni Tsetlin Machine Autoencoder (Omni TM-AE). All models were trained on the One Billion Word Benchmark corpus (Chelba et al., 2013) using a fixed vocabulary of 40,000 words. Word2Vec, FastText, and GloVe embeddings were configured with a vector dimension of 100, a window size of 5, and 25 training epochs (with GloVe using $MAX\_ITER$ = 25), while maintaining default hyperparameters elsewhere. The Omni TM-AE was trained using the following configuration: $number\_of\_examples$ = 2000, $accumulation$ = 24, $clauses$ = 32, $T$ = 20000, $s$ = 1.0, $epochs$ = 4, and $number\_of\_state\_bits\_ta$ = 8.

The experimental results (see Table 1) demonstrate the strong performance of the proposed method. The Omni TM-AE achieved the highest similarity scores on both the RG65 and MEN datasets (0.644 and 0.597, respectively), indicating its robustness across datasets of varying sizes. On average, Omni TM-AE attained the highest Kendall correlation (0.393). Although FastText achieved the highest Spearman correlation (0.550), the Omni TM-AE model followed closely with a score of 0.546—surpassing that of Word2Vec (0.545).

Embedding results for prior methods were excluded due to inherent limitations. The model from Bhattarai et al. (2024) was not scalable to larger vocabularies and could not generalize to tasks like classification and clustering. The model from Kadhim et al. (2025) required months of training and relied on a two-phase procedure, making its embeddings incompatible with clustering. For reference, RG65 was tested using TM-AE, which delivered lower performance than Omni TM-AE and required about 8 hours of training. In contrast, Omni TM-AE computes embeddings for the

Table 2: Comparison of accuracy using different embedding sources and classifier: LR (Logistic Regression), NB (Naive Bayes), RF (Random Forest), SVM (Support Vector Machine), MLP (Multi-layer Perceptron), and TM (Tsetlin Machine Classifier)

| Embedding Source | RF | LR | NB | SVM | MLP | TM | Average |
|---|---|---|---|---|---|---|---|
| GloVe | 0.81 | 0.83 | 0.82 | 0.80 | 0.82 | 0.80 | 0.813 |
| Word2Vec | 0.84 | 0.85 | 0.81 | 0.83 | 0.85 | 0.83 | 0.835 |
| FastText | 0.71 | 0.73 | 0.80 | 0.73 | 0.77 | 0.66 | 0.733 |
| Omni TM-AE | 0.83 | 0.85 | 0.81 | 0.82 | 0.84 | 0.83 | 0.830 |
| BERT | 0.80 | 0.84 | 0.81 | 0.81 | 0.82 | 0.79 | 0.811 |
| ELMo | 0.84 | 0.83 | 0.83 | 0.81 | 0.84 | 0.83 | 0.830 |

same dataset from the pre-generated embedding vectors takes only 47 seconds—approximately 600× faster—highlighting the scalability and reusability advantages of our approach.

## 4.3 CLASSIFICATION PERFORMANCE

To assess the effectiveness of the proposed embedding method in downstream tasks such as sentiment analysis, a series of classification experiments were conducted using multiple embedding and classification models. The embedding methods evaluated included GloVe, Word2Vec, Fast-Text, Omni TM-AE, BERT, and ELMo. For the classification step, six algorithms were utilized: Logistic Regression (LR), Naive Bayes (NB), Random Forest (RF), Support Vector Machine (SVM), Multi-layer Perceptron (MLP), and the TM Classifier.

All embedding models were trained using the IMDb dataset (Maas et al., 2011), with a vocabulary limit set to 20,000 tokens. The Word2Vec, FastText, and GloVe embeddings were configured with a vector size of 100, a window size of 5, and 25 training epochs ($MAX\_ITER$ = 25 in the case of GloVe), while all other hyperparameters remained at their default values. For BERT fine-tuning, the following settings were applied: a batch size of 8, a learning rate of 5e-5, three training epochs, and gradient clipping with a maximum norm of 1.0. The ELMo embeddings were obtained using the ELMo v3 model from TensorFlow Hub, which was used to generate token-level embeddings for pre-tokenized sentences using the model's default signature without applying any additional tokenization procedures. The configuration for Omni TM-AE was as follows: $number\_of\_examples$ = 2000, $accumulation$ = 24, $clauses$ = 32, $T$ = 20000, $s$ = 1.0, $epochs$ = 4, and $number\_of\_state\_bits\_ta$ = 8.

As part of the experiment, a perturbation strategy was implemented whereby 5% of the words in each document were substituted based on the document's sentiment label. For positively labeled documents, the five most similar words were retrieved using the embedding vectors, and one of these was randomly selected as a replacement. For negatively labeled documents, the 400 most similar words were retrieved. From this set, the five most semantically dissimilar words were selected, and one of these was then randomly chosen for substitution. This process aimed to evaluate the quality of semantic relationships captured by the embedding vectors in a sentiment classification context.

Subsequently, the modified documents were classified using the aforementioned classification models. RF was configured with $n\_estimators$=100. LR, SVM, MLP, and NB were used with their default configurations (only $random\_state$=42). For the TM, the following parameters were used uniformly across all embeddings: $clauses$=1000, $T$=8000, $s$=2.0, $epochs$=10, and $clause\_drop\_p$=0.75.

The classification results (see Table 2) reveal a strong performance by the proposed Omni TM-AE model, which achieved a high level of accuracy, ranking second overall across all classification algorithms. The accuracy margin separating Omni TM-AE from the top-performing model, Word2Vec, was minimal—with a very small difference of approximately half a percentage point. Specifically, Word2Vec achieved an accuracy score of 0.835, while Omni TM-AE closely followed with a score of 0.830. Notably, Omni TM-AE's performance was comparable to that of more complex models such as ELMo, which employs sentence-level embeddings, and it outperformed BERT (0.811), despite BERT's reliance on sophisticated transformer architectures. FastText, although an extension of

Table 3: Clustering performance comparison of the Omni TM-AE embedding model against baseline models (Word2Vec, FastText, GloVe) using five labeled datasets.

| Model Labels count Length | Scoring | 20 News 20 18,846 | Reuters 79 10,788 | Yelp 11,537 229,907 | Amazon 30 21,000 | AG News 5 120,001 | Average |
|---|---|---|---|---|---|---|---|
| Omni TM-AE | NMI | 0.2356 | 0.4402 | 0.5414 | 0.1229 | 0.5089 | 0.3698 |
|  | ARI | 0.1181 | 0.1102 | 0.0042 | 0.0348 | 0.5162 | 0.1567 |
| Word2Vec | NMI | 0.2558 | 0.4680 | 0.5494 | 0.1533 | 0.5145 | 0.3882 |
|  | ARI | 0.1179 | 0.1051 | 0.0056 | 0.0495 | 0.4893 | 0.1535 |
| Fast-Text | NMI | 0.2435 | 0.4718 | 0.5498 | 0.1523 | 0.5106 | 0.3856 |
|  | ARI | 0.1111 | 0.1196 | 0.0060 | 0.0470 | 0.4821 | 0.1532 |
| GloVe | NMI | 0.2342 | 0.4553 | 0.5498 | 0.1277 | 0.4760 | 0.3686 |
|  | ARI | 0.1154 | 0.0928 | 0.0050 | 0.0397 | 0.4500 | 0.1406 |

Word2Vec, demonstrated the lowest performance among the tested models with an accuracy of 0.733, while GloVe scored moderately at 0.813.

It was not feasible to include the model from Bhattarai et al. (2024) in this evaluation due to its limitations in scalability and reusability in generating embeddings. While the model from Kadhim et al. (2025) is theoretically applicable, its excessive training time—potentially extending over several months—renders it impractical within the scope of this study.

## 4.4 CLUSTERING ANALYSIS

This experiment constitutes the first implementation of NLP clustering application using the TM framework, which was not feasible with the models proposed in Works Bhattarai et al. (2024) and Kadhim et al. (2025). In this experiment, two methods are presented for evaluating embeddings produced by the proposed model. The first method involves a visual analysis using t-SNE (t-distributed stochastic neighbor embedding), a dimensionality reduction algorithm that is particularly useful for visualizing high-dimensional data.

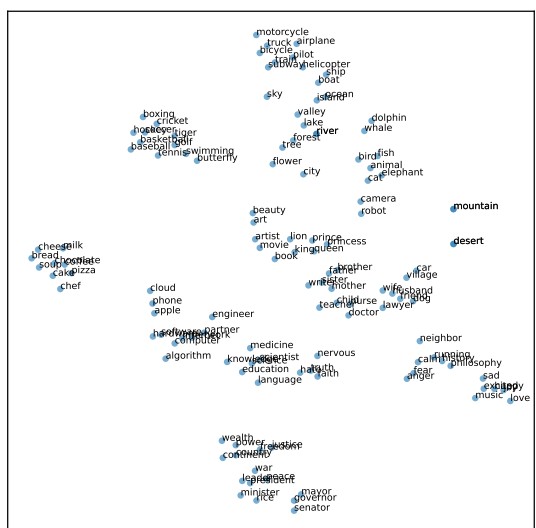

The model was trained to generate embeddings for 130 words, manually grouped into 13 semantic clusters, where each group contained words of similar meaning or context (e.g., the Technology group includes terms such as "computer," "Internet," and "robot"). Using the same experimental setup employed in the similarity experiment, the word embeddings were constructed. Figure 4 presents the t-SNE plot of the resulting embedding vectors. The t-SNE parameters used were: $n\_components$=2, $perplexity$=30, $learning\_rate$=200, $n\_iter$=2000, and $random\_state$=42.

Figure 4: t-SNE visualization of word embeddings generated by the Omni model for 130 words grouped into 13 semantic clusters.

The experiment demonstrated effective clustering. Distinct clusters were visually separable, while associative proximity was preserved between semantically adjacent groups. For instance, the word

"engineer" (positioned centrally and slightly toward the bottom-left in Figure 4), which belongs to the **Professions** cluster, appeared near the **Technology** cluster—an intuitive outcome given the semantic overlap between these fields. The word "partner" from **Relationships** cluster was also found in proximity to "engineer," prompting further analysis through the TM's interpretable capabilities.

To understand this positioning, an auxiliary experiment examined co-occurrence patterns of the word "partner" within the One Billion Word dataset (the source used for this t-SNE visualization). Results indicated that "partner" co-occurred with "network" in 552 documents and with "software" in 355 documents, whereas it co-occurred with "engineer" in only 29 documents. This justifies its embedding proximity to technology-related terms rather than strictly relationships-based ones. According to the TM-AE, the construction of the training variable $X$ relies on the presence of words in documents that support the target class being learned (as shown in Figure 2). This presence promotes a reward mechanism (memorization) during training, elevating the clause's literal state. Notably, the word frequency is not taken into account because the input vector $X$ is binary, indicating only the presence or absence of words, not their frequency.

Although this visualization experiment does not yield absolute accuracy, it offers an interpretable method for inferring semantic relationships, consistent with the transparent nature of the TM. This framework constructs learning logic through interpretable Boolean expressions rather than opaque mathematical operations (refer to the appendix for an example of TM transparency).

This initial experiment played a vital role in the hyperparameter tuning phase of the Omni TM-AE model, offering visual insights into embedding quality. However, such qualitative assessments may not provide robust quantitative evaluation metrics. To address this, a second experiment was conducted using document-level embeddings, where each document was represented as a vector by aggregating the word embeddings it contained. This allowed us to assess clustering performance at the document level, further demonstrating the flexibility of Omni TM-AE beyond word-level tasks. The embeddings for all models were generated using the same experimental conditions as described in the similarity experiment, using the One Billion Word dataset and a vocabulary of 40,000 tokens.

The evaluation relied on five public datasets containing documents with associated labels (News20, Reuters, Yelp, Amazon, and AG News), encompassing varying document lengths and label distributions. Clustering performance was evaluated using two standard metrics: Normalized Mutual Information (NMI) and Adjusted Rand Index (ARI), which assess clustering quality relative to ground truth labels. As shown in Table 3, the Omni TM-AE model achieved the highest ARI score (0.1567), outperforming Word2Vec (0.1535). In terms of NMI, Omni TM-AE attained 0.3698, a performance comparable to top models such as Word2Vec (0.3882) and FastText (0.3856), while GloVe lagged behind with a score of 0.3686.

## 5 CONCLUSION

This work presents a novel and interpretable embedding model, Omni TM-AE, based on the Tsetlin Machine Autoencoder. By integrating the information from all literals, including those traditionally excluded from clause formation, the proposed method constructs reusable and scalable embeddings using only a single training phase. This eliminates the retraining requirements found in earlier TM-based approaches and significantly reduces model complexity. Our experiments across semantic similarity, classification, and clustering tasks reveal that Omni TM-AE achieves strong performance compared to both classical embedding techniques and more complex contextual models like BERT and ELMo. Notably, it performs on par with or better than black-box models, while retaining a high level of interpretability—an essential trait for applications demanding transparency and diagnostic traceability. In summary, Omni TM-AE marks a substantial advancement in bridging the gap between interpretable learning and high-performing NLP embeddings, laying the groundwork for more responsible and comprehensible AI systems.

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

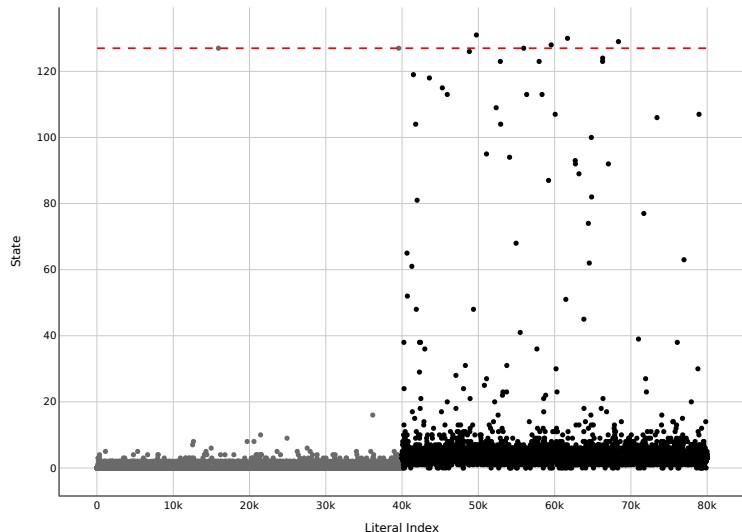

Figure 5: Distribution of literal after training with a large vocabulary of 40,000 tokens (80,000 literals including negations). The model learns to reduce literal states for original tokens to improve clause discrimination within a few epochs.

## 6    APPENDIX

### 6.1    OMNI TM-AE EMBEDDING STRUCTURE

A clause containing its original literals (indexed from 1 to 40,000 along the x-axis) as illustrated in Figure 5 presents certain challenges in effectively utilizing their state values for embedding computation, as indicated by the concentration of gray dots within a narrow state range (0 to 25). This limited range of the original literals' state values becomes more pronounced when the vocabulary size is expanded—from 1,600 tokens, as shown in Figure 3, to 40,000 tokens in Figure 5. Despite this limitation, increasing the vocabulary size offers the advantage of enabling more rapid convergence of the TM during the training process.

To overcome the constraint associated with the restricted state range of original literals, the proposed method computes the embedding by incorporating the state values of both the original literals and their negations (negated literals ranging from 40,001 to 80,000 in Figure 5). This inclusive strategy allows the model to leverage the full extent of the clause structure—hence the term Omni.

Compared to previous approaches, the proposed method introduces a novel framework for generating scalable and reusable embeddings—an aspect that was not effectively achieved in the work Bhattarai et al. (2024). Moreover, it delivers a highly accurate and efficient embedding mechanism, while eliminating the complexity associated with multi-phase training procedures, as required in the work Kadhim et al. (2025).

To provide a clear description of the proposed embedding computation process used in the Omni TM-AE model, Algorithm 1 presents the detailed steps required to construct the embedding vector for a target word from the clause-level literal states. The algorithm illustrates how the signed state values of both the original literals and their negations are aggregated, normalized, and combined into the final embedding vector. This formalization highlights the simplicity and efficiency of the proposed method, which eliminates the need for multi-phase training or complex post-processing steps.

### 6.2    OMNI TM-AE INTERPRETABILITY

Our approach preserves the core principles of the TM, particularly its inherent transparency and interpretability. As demonstrated in Figure 3, the resulting embeddings are readily diagnosable and analytically traceable, allowing for a clear understanding of the underlying training dynamics.

---

**Algorithm 1** Embedding Vector Computation for a Target Word Using Positive Clauses

---

**Require:** Set of positive-weighted clauses $\mathcal{C}$ for the target word, where each clause $c \in \mathcal{C}$ contains $S(c)$ set of literal-state pairs $(l, n)$, Vocabulary $\mathcal{D} = \{x_1, x_2, \ldots, x_d\}$
**Ensure:** Embedding vector $\mathbf{e} \in \mathbb{Z}^d$
 1: **Function** COMPUTEEMBEDDING$(\mathcal{C}, \mathcal{D})$
 2: Initialize $\mathbf{v} \leftarrow \mathbf{0}_d$                                    $\triangleright$ Signed state sums
 3: $t \leftarrow |\mathcal{C}|$                               $\triangleright$ Total number of positive clauses
 4: **for** $i \leftarrow 1$ **to** $d$ **do**
 5:     **for all** clause $c \in \mathcal{C}$ **do**
 6:         **for all** $(l, n) \in S(c)$ **do**
 7:             **if** $l$ is of the form $x_i$ **then**
 8:                 $v_i \leftarrow v_i + n$
 9:             **else if** $l$ is of the form $\neg x_i$ **then**
10:                 $v_i \leftarrow v_i - n$
11:             **end if**
12:         **end for**
13:     **end for**
14:     $e_i \leftarrow \left\lfloor \frac{v_i}{t} \right\rfloor$
15: **end for**
16: **return** $\mathbf{e} = [e_1, e_2, \ldots, e_d]$

---

The associations formed between the target class and the remaining vocabulary are transparent and straightforward to interpret. This model thus presents a significant advancement in providing interpretable and analytically robust embeddings compared to existing embedding methods.

To further illustrate the interpretability of the proposed method, consider the words "driving" and "road" as examples. After embedding and training using the Omni TM-AE approach, it is possible to extract the clauses that represent the relationship between each target word and the other features in the vocabulary. For instance, in the case of the target word "driving," the clauses indicate that related words such as "vehicle," "license," "road," and "safe" attain high state values, highlighting strong contextual relationships. Similarly, for the target word "road," the clauses reveal that words like "vehicle," "smooth," "driver," and "traffic" also maintain high state values.

Moreover, if we seek to trace why certain words, such as "vehicle," exhibit similar contributions in the embeddings of both "driving" and "road," this can be directly accomplished by independently analyzing the training stages of the TM for each target word. Given that all stages involved in the embedding construction process, as illustrated in Figure 2, rely on simple, transparent operations—such as reward and penalty-based feedback updates, thresholding and AND operations in the evaluation stage, and accumulation during the preparation stage—it becomes feasible to systematically infer and explain the mechanisms leading to such shared associations. Unlike neural network-based methods, which often involve complex and opaque learning processes, the proposed approach ensures that all steps are interpretable, traceable, and analytically accessible.

