# OpenReview forum: "Omni TM-AE: A Scalable and Interpretable Embedding Model Using the Full Tsetlin Machine State Space"
_ICLR.cc/2026/Conference — ICLR 2026 Conference Withdrawn Submission_

### Official Review · Reviewer_iBFj · 2025-10-23

**Soundness:** 2
**Presentation:** 2
**Contribution:** 1
**Rating:** 2
**Confidence:** 4

**Summary:**

This paper proposes Omni TM-AE, a variant of Tsetlin Machine–based autoencoder models that constructs word-level embedding vectors by aggregating both active and previously excluded literals through a signed aggregation mechanism. The authors argue this design enhances interpretability while maintaining competitive embedding quality. The model is assessed on several word similarity datasets and downstream document classification and clustering tasks by averaging word embeddings to form sentence or document representations. The results are compared to standard static baselines such as Word2Vec, GloVe, and FastText.

**Strengths:**

1. The design encourages traceability: because each dimension corresponds directly to literal states, the model allows users to track how reward and penalty signals contribute to the final embedding. This kind of mechanistic interpretability is not common in standard neural embedding approaches.

2. The empirical coverage is reasonably broad, spanning similarity, clustering, and classification settings. The inclusion of implementation details and code facilitates reproducibility, even if the benchmark scope itself is somewhat limited.

**Weaknesses:**

1. The proposed approach is limited to word-level representations. Sentence and document vectors are derived through simple averaging, a composition strategy that lags behind the richer contextual embedding techniques now standard in practice (e.g., SBERT-like models). Importantly, the method is not evaluated on the MTEB benchmark, which has become the principal standard for embedding model comparisons across retrieval, STS, reranking, clustering, and classification tasks. This omission weakens the paper’s empirical positioning.

2. Baseline selection is narrow. The paper focuses on static embeddings and a few TM variants but omits widely recognized interpretable or sparse embedding methods (such as SPINE, Word2Sense, ultradense embeddings, or analytical approaches). Without these, it’s difficult to meaningfully situate the work in the interpretability space or quantify its comparative advantages.

3. Although the approach provides literal-level transparency, it largely encodes bag-of-words associations rather than contextual or conceptual meaning. It lacks the semantic structure that other interpretable embedding frameworks attempt to model, limiting its explanatory depth.

4. The contribution is incremental rather than transformative: incorporating excluded literals and simplifying the training pipeline extends the TM-AE line but doesn’t constitute a conceptual shift in representation learning. The paper would benefit from a stronger articulation of why this specific modification matters in a broader embedding context.

**Questions:**

1. Do the authors intend to explore more expressive sentence-level composition functions or benchmark against MTEB to make the work more competitive with contemporary embedding methods?

2. Why were interpretable or sparse embedding baselines omitted? Including SPINE, Word2Sense, or ultradense methods would provide a more complete interpretability comparison.

3. How might the proposed approach be adapted to move beyond literal-level signals toward richer, concept-level semantics?

4. How do the authors view their contribution in terms of novelty relative to the prior TM-AE work, and how might this design scale to more competitive benchmarks?

---

### Official Review · Reviewer_JjEy · 2025-10-29

**Soundness:** 1
**Presentation:** 1
**Contribution:** 1
**Rating:** 0
**Confidence:** 5

**Summary:**

Desk reject due to author names appearing in the title page and not following ICLR style.

**Strengths:**

I have not conducted a technical review of this paper because of its style and anonymity violations.

**Weaknesses:**

I have not conducted a technical review of this paper because of its style and anonymity violations.

**Questions:**

I have not conducted a technical review of this paper because of its style and anonymity violations.

---

### Official Review · Reviewer_1C8W · 2025-11-01

**Soundness:** 1
**Presentation:** 2
**Contribution:** 2
**Rating:** 2
**Confidence:** 4

**Summary:**

This paper introduces Omni TM-AE, an interpretable embedding model based on the Tsetlin Machine framework. Unlike previous TM-based autoencoders, Omni TM-AE fully uses the TM’s entire state matrix. This design allows the model to capture richer contextual information and produce embeddings in a single training phase. The authors evaluate Omni TM-AE across semantic similarity, sentiment classification, and document clustering tasks, showing competitive results compared to established embedding baselines such as Word2Vec, FastText, and GloVe. The method retains the transparency of TM-based reasoning while improving scalability and reusability.

**Strengths:**

- Leveraging the full TM state space (including excluded literals) is conceptually innovative and practically impactful, addressing both interpretability and reusability.
- Results across diverse datasets show that Omni TM-AE can exceed traditional embeddings, particularly in tasks involving semantic similarity.

**Weaknesses:**

### W1. Limited task coverage
The method produces only word-level embeddings, with sentence/document representations obtained by simple averaging. It's shallow compared to modern contextual or sentence-level embeddings (e.g., SBERT, SimCSE) used in real-world retrieval and classification pipelines. Extending experiments to sentence-level benchmarks (e.g., MTEB) would clarify scalability and real-world utility.

### W2. Missing interpretability baselines.
Comparisons are limited to black-box and prior TM variants. The paper omits interpretable or sparse embedding baselines (e.g., sparse autoencoders, dictionary-based or concept-factor models), weakening claims of interpretability and generality.

### W3. Mechanistic rather than semantic interpretability
Interpretability is confined to tracing literal states rather than uncovering human-understandable semantic features. No quantitative or human-centered evaluation supports the interpretability claim.

### W4. Incremental contribution within existing TM-AE models
The main technical novelty, folding excluded literals into embeddings through signed state aggregation, extends the earlier TM-AE work but remains a modest refinement.

### W5. Missing ablation and sensitivity analyses
The paper lacks ablations isolating the contribution of excluded literals and does not discuss sensitivity to key hyperparameters such as clause number or threshold.

**Questions:**

- Can you quantify how excluded literals improve embedding quality (e.g., via ablation)?

- How sensitive is the model to hyperparameters such as clause count and threshold?

- Could user studies verify whether the model’s literal-based associations align with human-understandable semantics?

- How does Omni TM-AE generalize to contextual or sentence-level embedding tasks? Could the single-phase mechanism scale to larger or dynamic corpora?

- How does it compare to interpretable embedding approaches in performance and interpretability trade-offs?

---

### Note · Authors · 2025-11-17

**Comment:**

We respectfully request the withdrawal of our submission. After discussing this matter with all co-authors, we collectively decided to retract the paper at this time. Thank you for your consideration.

**Withdrawal Confirmation:**

I have read and agree with the venue's withdrawal policy on behalf of myself and my co-authors.